# microGAN: Promoting Variety through Micro-batch Discrimination

## Abstract

We propose to tackle the mode collapse problem in generative adversarial networks (GANs) by using multiple discriminators and assigning a different portion of each minibatch, called microbatch, to each discriminator. We gradually change each discriminator's task from distinguishing between real and fake samples to discriminating samples coming from inside or outside its assigned microbatch by using a diversity parameter $\alpha$. The generator is then forced to promote variety in each minibatch to make the microbatch discrimination harder to achieve by each discriminator. Thus, all models in our framework benefit from having variety in the generated set to reduce their respective losses. We show evidence that our solution promotes sample diversity since early training stages on multiple datasets.

## 1 Introduction

Generative adversarial networks (Goodfellow et al. (2014)), or GANs, consist of a framework describing the interaction between two different models - one generator (G) and one discriminator (D) - that are trained together. While $G$ tries to learn the real data distribution by generating realistic looking samples that are able to fool $D$, $D$ tries to do a better job at distinguishing between real and the fake samples produced by $G$. Although showing very promising results across various domains (Edwards and Storkey (2015); Ho and Ermon (2016); Yu et al. (2017); Kim et al. (2017a); Yang et al. (2017); Donahue et al. (2018)) , GANs have also been continually associated with instability in training, more specifically mode collapse (Kim et al. (2017b); Arjovsky and Bottou (2017); Mescheder et al. (2017); Che et al. (2016); Arjovsky et al. (2017)). This behavior is observed when $G$ is able to fool $D$ by only generating samples from the same data mode, leading to very similar looking generated samples. This suggests that $G$ did not succeed in learning the full data distribution but, instead, only a small part of it. This is the main problem we are trying to solve with this work.

The proposed solution is to use multiple discriminators and assign each $D$ a different portion of the real and fake minibatches, *i.e.,* microbatch. Then, we update each $D$'s task to discriminate between samples coming from its assigned fake microbatch and samples from the microbatches assigned to the other discriminators, together with the real samples. We call this microbatch discrimination. Throughout training, we gradually change from the originally proposed real and fake discrimination by Goodfellow et al. (2014) to the introduced microbatch discrimination by the use of an additional diversity parameter $\alpha$ that ultimately controls the diversity in the overall minibatch.

The main idea of this work is to force $G$ to reduce its loss by inducing variety in the generated set, complicating each $D$'s task on separating the samples in its microbatch from the rest. Even though only producing very similar images would also complicate the desired discrimination, it would not benefit any of the models. This is due to the attribution of distinct probabilities by each $D$ to samples from and outside its microbatch being required to minimize $G$ and $D$'s losses. Hence, all models in the proposed framework, called microGAN, benefit directly from diversity in the generated set.

Our main contributions can be stated as follows: (i) proposal of a novel multi-adversarial GANs framework[1] (Section 3) that mitigates the inherent mode collapse problem in GANs; (ii) empirical evidence on multiple datasets showing the success of our approach in promoting sample variety since early stages of training (Section 4) (iii) Competitiveness against other previously proposed methods on multiple datasets and evaluation metrics (Section 5).

---

[1]Code will be available upon publication.

## 1.1 RELATED WORK

Previous works have optimized GANs training by changing the overall models' objectives, either by using discrepancy measurements (Li et al. (2017); Sutherland et al. (2016)) or different divergence functions (Nowozin et al. (2016); Uehara et al. (2016)) to approximate the real data distribution. Moreover, Zhao et al. (2016); Berthelot et al. (2017); Unterthiner et al. (2017) proposed to use energy-driven objective functions to encourage sample variety, Mroueh et al. (2017) tried to match the mean and covariance of the real data, and Metz et al. (2016) used an unrolled optimization of $D$ to train $G$. Che et al. (2016); Warde-Farley and Bengio (2016); Wang et al. (2017); Berthelot et al. (2017) penalized missing modes by using an extra autoenconder in the framework. Salimans et al. (2016) performed minibatch discrimination by forcing $D$ to condition its output on the similarity between the samples in the minibatch. Springenberg (2015) increased $D$'s robustness by maximizing the mutual information between inputs and corresponding labels, while Lin et al. (2017) forced $D$ to make decisions over multiple samples of the same class, instead of independently.

Regarding using multiple discriminators, Neyshabur et al. (2017) extended the framework to several discriminators with each focusing in a low-dimensional projection of the data, set a priori. Durugkar et al. (2016) proposed GMAN, consisting of an ensemble of discriminators that could be accessed by the single generator according to different levels of difficulty. Nguyen et al. (2017) introduced D2GAN, introducing a single generator dual discriminator architecture where one discriminator rewards samples coming from the true data distribution whilst the other rewards samples coming from the generator, forcing the generator to continuously change its output. Mordido et al. (2018) applied adversarial dropout by omitting the feedback of a given $D$ at the end of each batch.

## 2 GENERATIVE ADVERSARIAL NETWORKS

The original GANs framework (Goodfellow et al. (2014)) consists of two models: a generator ($G$) and a discriminator ($D$). Both models are assigned different tasks: whilst $G$ tries to capture the real data distribution $p_r$, $D$ learns how to distinguish real from fake samples. $G$ maps a noise vector $z$, retrieved from a noise distribution $p_z$, to a realistic looking sample belonging to the data space. $D$ maps a sample to a probability $p$, representing the likeliness of that given sample coming from $p_r$ rather than from $p_g$. The two models are trained together and play the following minimax game:

$$\min_G \max_D V(D, G) = \mathbb{E}_{x \sim p_r(x)}[\log D(x)] + \mathbb{E}_{z \sim p_z(z)}[\log(1 - D(G(z)))], \tag{1}$$

where $D$ maximizes the probability of assigning samples to the correct distribution and $G$ minimizes the probability of its samples being considered from the fake data distribution.

Alternatively, one can also train $G$ to maximize the probability of its output being considered from the real data distribution, *i.e.,* $\log D(G(z))$. Even though this changes the type of the game, by being no longer minimax, it avoids the saturation of the gradient signals at the beginning of training (Goodfellow et al. (2014)), where $G$ only receives continuously negative feedback, making training more stable in practice. However, since we employ multiple discriminators in the proposed framework, it is less likely that $G$ does not receive any positive feedback from the whole adversarial ensemble (Durugkar et al. (2016)). Therefore, we make use of the original value function in this work.

## 3 MICROGAN

In this work, we propose a novel generative multi-adversarial framework named microGAN, where we start by splitting each minibatch into several microbatches and assigning a unique one to each $D$. The key aspect of this work is the usage of microbatch discrimination, where we change the original discrimination task of distinguishing between real and fake samples, as proposed in Goodfellow et al. (2014), to each $D$ distinguishing between samples coming or not from its fake microbatch. This change is performed in a gradual fashion, using an additional diversity parameter $\alpha$. Thus, each $D$'s output gradually changes from the probability of a given sample being real to the probability of a given sample not belonging to its fake microbatch. Moreover, since each $D$ is trained with different fake and real samples, we encourage them to focus on different data properties. Figure 1 illustrates the proposed framework.

The proposed microbatch-level discrimination task leads to $G$ making such discrimination harder for each $D$ to lower its loss. Hence, $G$ is forced to induce variety on the overall minibatch, making it a substantially harder task for each $D$ to be able to separate its subset of fake samples in the diverse minibatch. Note that producing very similar samples across the whole minibatch would also make such discrimination difficult by making the whole minibatch the same. However, $G$ also benefits from each $D$ assigning distinct probabilities to samples from inside and outside its designed microbatch to lower its loss, making the generation of different samples in the minibatch a necessary requirement to obtain different outputs from $D$. Hence, all models in our framework benefit directly from sample variety in the generated set.

In the microGAN scenario with a positive diversity parameter $\alpha$, each $D$ assigns low probabilities to fake samples from its microbatch and high probabilities to fake samples from the rest of the microbatches as well as samples from the real data distribution. Hence, fake samples in the rest of the minibatch, *i.e.,* not coming from its assigned microbatch, shall be given distinct output probabilities by each $D$. On the other hand, $G$ minimizes the probability given by each $D$ to the samples outside its microbatch

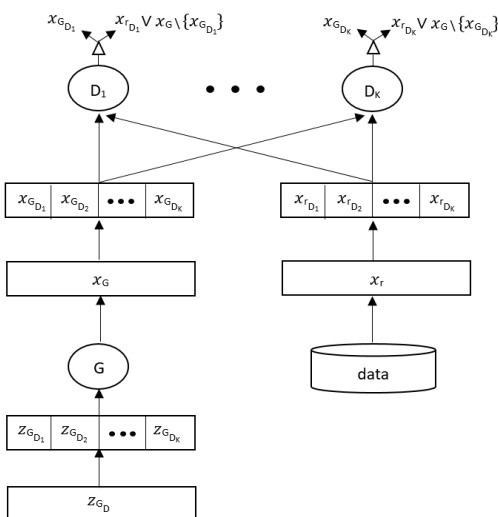

Figure 1: microGAN framework assuming a positive diversity parameter $\alpha$. Each discriminator $D_k$ is assigned a different microbatch $x_{G_{D_k}}$, where it discriminates between samples coming from inside its microbatch and samples coming from the microbatches assigned to the rest of the discriminators ($x_G \backslash x_{G_{D_k}}$) together the real samples $x_{r_{D_k}}$.

and maximizes the probability given to the fake samples assigned to that specific $D$. The value function of our minimax game is as follows:

$$\min_G \max_{\left\{D_k\right\}} \sum_{k=1}^{K} V(D_k, G) = \sum_{k=1}^{K} \mathbb{E}_{x \sim p_{r_{D_k}}(x)}[\log D_k(x)] + \mathbb{E}_{z \sim p_{z_{G_{D_k}}}(z)}[\log(1 - D_k(G(z)))]$$
$$+ \alpha \times \mathbb{E}_{z' \sim p_{z_{G_D} \backslash \{z_{G_{D_k}}\}}(z')}[\log D_k(G(z'))], \tag{2}$$

where K represents the number of total discriminators in the set. $p_{r_{D_k}}$ represents real samples from $D_k$'s real microbatch, $p_{z_{G_{D_k}}}$ indicates fake samples from $D_k$'s fake microbatch, and $p_{z_{G_D} \backslash \{z_{G_{D_k}}\}}$ relates to the rest of the fake samples in the minibatch but not in $p_{z_{G_{D_k}}}$. $\alpha$ represents the *diversity parameter* responsible for penalizing the incorrect discrimination of fake samples coming from $p_{z_{G_D} \backslash \{z_{G_{D_k}}\}}$ by each $D_k$. Note that $\alpha = 0$ would represent the original GANs objective for each $D$ in the set. Appropriate $\alpha$ values and its effects on the whole minibatch diversity are discussed next.

## 3.1 DIVERSITY PARAMETER $\alpha$

We control the weight of the microbatch discrimination in the models' losses by introducing an additional diversity parameter $\alpha$. Lower $\alpha$ values lead to $G$ significantly lowering its loss by generating realistic looking samples on each microbatch without taking much consideration on the variety of the overall minibatch. On the other hand, higher $\alpha$ values induce a stronger effect on $G$'s loss if each $D$ is able to discriminate between samples inside and outside its microbatch. However, high values of $\alpha$ might compromise the realistic properties of the produced samples, since too much weight is given to the last part of Eq. 2, being sufficient to effectively minimize $G$'s loss. Thus, using $\alpha > 0$ represents an additional way of ensuring data variety within the minibatch produced by $G$ at each iteration. An overview of different possible $\alpha$ settings follows below.

### 3.1.1 STATIC $\alpha$

First, we statically set $\alpha$ to values between 0 and 1 throughout the whole training. For the evaluation of the effects of each $\alpha$ value, we used a toy experiment of a 2D mixture of 8 Gaussian distributions (representing 8 data modes) firstly presented by Metz et al. (2016), and further adopted by Nguyen et al. (2017). We used 8 discriminators for all the experiments. Results are shown in Figure 2.

When setting $\alpha = 0$, $G$ mode collapses on a specific mode, showing the importance of using positive $\alpha$ values to mitigate mode collapse. When setting $0.1 \leq \alpha \leq 0.5$, $G$ is able to capture all data modes during training. However, learning problems in the early stages are observed, with $G$ only focusing on promoting variety in the generated samples. For higher $\alpha$ values ($\alpha \geq 0.6$), $G$ was unable to produce any realistic looking samples throughout the whole training, focusing solely on sample diversity to lower its loss, suggesting the dominance of the last part of Eq. 2. Hence, a mild, dynamic, manipulation of $\alpha$ values seems to be necessary for a successful training of $G$, ultimately meaning both realistic and diverse samples from an early training stage.

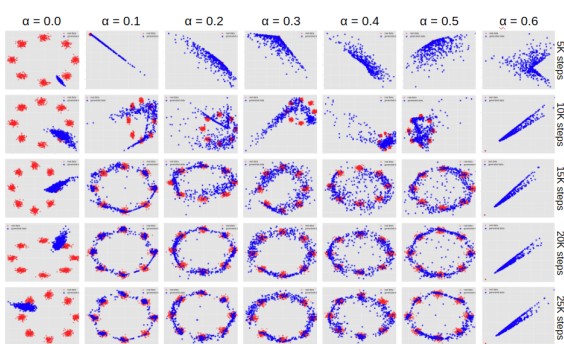

Figure 2: Toy experiment using static $\alpha$ values. Real data is presented in red while generated data is in blue.

### 3.1.2 SELF-LEARNED $\alpha$

We dynamically set $\alpha$ over time by adding it as a parameter of $G$ and letting it self-learn its values to lower its loss. However, we observed that $G$ takes advantage of being able to reduce its loss by increasing $\alpha$ at a large rate, focusing simply on promoting diversity in the generated samples without much realism, similarly to what was observed when using $\alpha = 0.6$ in the toy experiment (Figure 2). Hence, we suggest several properties that $\alpha$ should have so that diversity does not compromise the veracity of the generated samples.

First, $\alpha$ should be upper bounded so that the last part of Eq. 2 (responsible for sample diversity) does not overpower the first part (responsible for sample realism), ultimately not compromising the feedback given to $G$ to also be able to generate realistic samples. Second, $\alpha$'s growth should saturate over time, meaning that continuously increasing at large rates $\alpha$ is no longer an option to substantially decrease $G$'s loss over time. Lastly, to tackle the problem in learning of early to mid stages, we suggest that $\alpha$ should grow in a controlled fashion, so focus can also be given in the realistic aspect of the samples since the beginning of training.

Thus, we propose to make $\alpha$ a function of $\beta$, where $\alpha(\beta) \in [0, 1[$, and let $G$ regulate $\beta$ instead of directly learning $\alpha$. We evaluated regulating $\alpha$ over three different functions that have the desired properties:

$$\alpha(\beta) = \begin{cases} \alpha_{sigm}(\beta) = Sigmoid(\beta), \beta \geq \beta_{sigm} \\ \alpha_{soft}(\beta) = Softsign(\beta), \beta \geq \beta_{soft} \\ \alpha_{tanh}(\beta) = Tanh(\beta), \beta \geq \beta_{tanh} \end{cases} \tag{3}$$

with $\beta_{sigm}, \beta_{soft}$, and $\beta_{tanh}$ representing the initial values of $\beta$ when training begins for the respective functions. For all the experiments of this paper, we set $\beta_{tanh} = \beta_{soft} = 0$, to obtain a positive codomain, and $\beta_{sigm} = -1.8$, since we achieved better empirical results by starting $\beta$ with this value (for further discussion about the effects of using different $\beta_{sigm}$ on $\alpha_{sigm}(\beta)$'s growth please see the Appendix). Note that learning $\alpha$ without any constraints can be characterized as using the identity function ($\alpha(\beta) = \alpha_{ident}(\beta) = \beta$). Thus, each used function promotes a different $\alpha$ growth over time. To ease presentation, we neglect to write $\beta$'s dependence for the rest of the manuscript and use only the function names to described each $\alpha$ setting: $\alpha_{sigm}, \alpha_{soft}, \alpha_{tanh}$, and $\alpha_{ident}$.

Results on the toy dataset using the different proposed $\alpha$ functions are shown in Figure 3. The benefits of increasing $\alpha$ in a milder fashion, as performed when using $\alpha_{sigm}$, are observed especially early on training, with $G$ being concerned with the realism of the generated samples. On the other hand, when using $\alpha_{tanh}$ and $\alpha_{soft}$, the network takes longer to focus on the data realism (10K steps) since it is able to reduce its loss significantly by simply promoting variety due to the steeper growth of $\alpha$ in the earlier stages on both functions. Nevertheless, as the functions gradually saturate, all $\alpha$ settings manage to eventually capture the real data distribution while still keeping the diversity in the generated samples.

In conclusion, one can summarize micro-GAN's training using these variations of self-learned $\alpha$ as the following: in the first iterations, $G$ increases $\alpha$ to reduce its loss, expanding its output. As $\alpha$ starts to saturate and each $D$ learns how to distinguish between real and fake samples, $G$ is forced to lower its loss by creating both realistic and diverse samples.

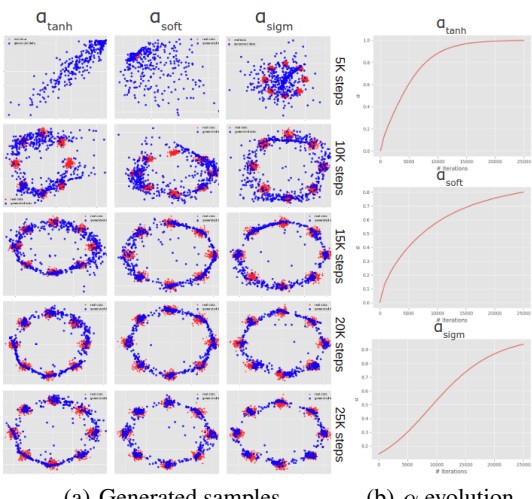

(a) Generated samples.  (b) $\alpha$ evolution.

Figure 3: Analysis of using different $\alpha$ functions on the toy dataset. The generated samples are shown in (a). The evolution of $\alpha$ on each function is presented in (b).

# 4 EXPERIMENTAL RESULTS

We validated the effects of using different $\alpha$ functions on MNIST (LeCun and Cortes (2010)), CIFAR-10 (Krizhevsky (2009)), and cropped CelebA (Liu et al. (2015)). To quantitatively evaluate such effects, we used the Fréchet Inception Distance (Heusel et al. (2017)), or FID, since it has been shown to be sensitive to image quality as well as mode collapse (Lucic et al. (2017)), with the returned distance increasing notably when modes are missing from the generated data. We used several variations of the standard FID for a thorough study of $\alpha$'s effects in training, as well as the influence of using a different number of discriminators in our framework.

## 4.1 INTRA FID

To measure the variety of samples of the generated set, we propose to calculate the FID between two subsets of 10K randomly picked fake samples generated at the end of every thousand iterations. We call this metric Intra FID. Important to note that Intra FID only measures the diversity in the generated set, not its realism. Hence, higher values indicate more diversity within the generated samples while lower values might indicate mode collapse in the generated set. The relation between Intra FID and progressive values of $\alpha$ is shown in Figure 4.

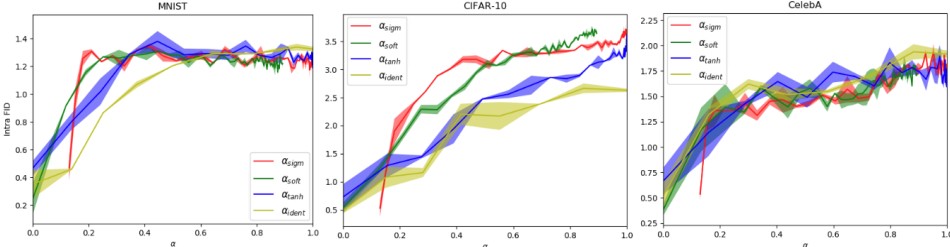

Figure 4: Intra FID as $\alpha$ progresses. Higher values represent higher variety in the generated set.

We observe a strong correlation between $\alpha$'s growth and variety in the set, especially in beginning to mid-training. Later on, as $\alpha$ saturates, the variety is kept (represented by the stability of the Intra

FID). It is further visible that $\alpha_{sigm}$, $\alpha_{soft}$, and $\alpha_{tanh}$ converge to similar Intra FID on all datasets. Important to note, that, to ease the visualization, the graphs only represent $0 \leq \alpha \leq 1$, with $\alpha_{ident}$'s values naturally surpassing 1 as time progresses.

## 4.2 CUMULATIVE INTRA FID

To analyze the sample variety over time, we summed the Intra FID values obtained from every thousand iterations. Hence, higher values indicate that the model was able to promote more variety in the set across time. Results are shown in Figure 5, where we observe that using more discriminators leads to more variety across all datasets and $\alpha$ functions. Moreover, using $\alpha = 0$ leads to lower variety compared to using positive $\alpha$ values, with $\alpha_{sigm}$, $\alpha_{soft}$, and $\alpha_{tanh}$ obtaining similar values throughout the different datasets. Even though $\alpha_{ident}$ promotes the highest variety, the generated samples lack realism, as previously witnessed in the toy experiment and further discussed next.

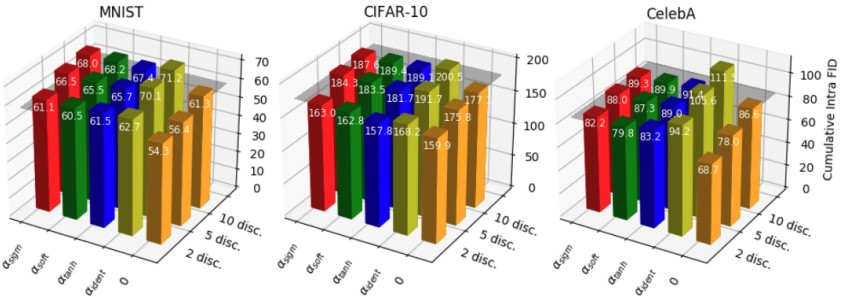

Figure 5: Cumulative Intra FID using a different number of discriminators and $\alpha$ functions on the different datasets. Higher values correlate to higher variety in the produced samples across time. Values obtained using standard GANs are represented by the grey plane as a baseline.

## 4.3 MEAN AND MINIMUM FID

To analyze both the realism and variety of the generated samples, we used the standard FID calculated between 10K fake samples and the real training data. Lower values should indicate both diversity and high-quality samples. The Mean FID and Minimum FID across 50K iterations are presented in Table 1 for each dataset. We observe that the best values, both in terms of mean and minimum, are obtained when using a higher number of discriminators, *i.e.,* 5 or 10, and $\alpha_{tanh}$, $\alpha_{soft}$, and $\alpha_{sigm}$. Moreover, the high distances obtained when using $\alpha_{ident}$ confirm the lack of realism of the generated samples, highlighting the importance of constraining $\alpha$ by the properties previously stated in Section 3.

Table 1: Mean and Minimum FID over 50K iterations on the different datasets.

| MICROGAN | | MNIST | | CIFAR-10 | | CELEBA | |
|---|---|---|---|---|---|---|---|
| K | $\alpha$ | MEAN FID | MIN FID | MEAN FID | MIN FID | MEAN FID | MIN FID |
| 1 | - | $50.9 \pm 9.7$ | $22.7 \pm 0.7$ | $125.5 \pm 1.5$ | $84.8 \pm 1.6$ | $77.3 \pm 1.7$ | $38.5 \pm 1.1$ |
| 2 | $\alpha_{sigm}$ | $37.6 \pm 1.1$ | $23.5 \pm 3.0$ | $111.9 \pm 0.1$ | $90.8 \pm 0.6$ | $76.3 \pm 0.6$ | $53.0 \pm 2.6$ |
| 2 | $\alpha_{soft}$ | $41.9 \pm 1.2$ | $24.6 \pm 0.0$ | $110.2 \pm 0.9$ | $90.6 \pm 1.2$ | $74.7 \pm 2.9$ | $49.5 \pm 0.1$ |
| 2 | $\alpha_{tanh}$ | $43.9 \pm 0.8$ | $27.2 \pm 0.5$ | $115.3 \pm 0.5$ | $91.3 \pm 0.4$ | $87.1 \pm 2.4$ | $54.7 \pm 0.8$ |
| 2 | $\alpha_{ident}$ | $89.1 \pm 2.2$ | $53.6 \pm 2.9$ | $168.1 \pm 2.0$ | $113.2 \pm 2.2$ | $206.1 \pm 3.5$ | $113.6 \pm 5.2$ |
| 5 | $\alpha_{sigm}$ | $\mathbf{34.7 \pm 0.3}$ | $20.1 \pm 0.1$ | $\mathbf{103.9 \pm 1.8}$ | $81.4 \pm 1.1$ | $\mathbf{66.5 \pm 0.6}$ | $40.4 \pm 3.1$ |
| 5 | $\alpha_{soft}$ | $37.2 \pm 0.3$ | $\underline{19.4 \pm 0.1}$ | $106.4 \pm 0.8$ | $82.5 \pm 1.2$ | $69.1 \pm 0.3$ | $42.0 \pm 2.0$ |
| 5 | $\alpha_{tanh}$ | $39.4 \pm 1.1$ | $20.0 \pm 0.1$ | $107.2 \pm 0.8$ | $\mathbf{80.8 \pm 0.6}$ | $70.3 \pm 1.3$ | $42.8 \pm 0.5$ |
| 5 | $\alpha_{ident}$ | $61.2 \pm 0.3$ | $37.3 \pm 0.2$ | $127.9 \pm 0.4$ | $97.5 \pm 2.8$ | $135.9 \pm 1.1$ | $77.5 \pm 2.0$ |
| 10 | $\alpha_{sigm}$ | $38.9 \pm 3.0$ | $18.0 \pm 0.1$ | $110.2 \pm 1.7$ | $79.0 \pm 0.7$ | $68.4 \pm 0.1$ | $34.8 \pm 1.2$ |
| 10 | $\alpha_{soft}$ | $\underline{36.2 \pm 0.9}$ | $\mathbf{17.1 \pm 0.2}$ | $110.8 \pm 0.4$ | $79.2 \pm 0.5$ | $\underline{67.8 \pm 2.6}$ | $\mathbf{34.5 \pm 0.2}$ |
| 10 | $\alpha_{tanh}$ | $37.4 \pm 1.2$ | $17.4 \pm 0.2$ | $112.8 \pm 1.7$ | $\mathbf{77.7 \pm 0.6}$ | $71.0 \pm 1.4$ | $34.5 \pm 0.3$ |
| 10 | $\alpha_{ident}$ | $48.7 \pm 0.9$ | $28.7 \pm 0.1$ | $117.0 \pm 0.2$ | $87.1 \pm 1.0$ | $91.4 \pm 0.2$ | $45.4 \pm 0.1$ |

## 4.4 GENERATED SAMPLES

The generated samples on each dataset using 1 and 10 discriminators with different $\alpha$ are presented in Figure 6. For an objective assessment of the variety by the end of each iteration, the Intra FID is also provided. We observe the superiority of the generated samples, both in terms of realism and variety, when using $\alpha_{sigm}$, $\alpha_{soft}$, and $\alpha_{tanh}$ on all datasets. However, $\alpha_{tanh}$ seems to show a delayed ability in generating realistic samples, possibly due to the increase of $\alpha$ at a steeper fashion. The inability of generating realistic samples when using $\alpha_{ident}$ is also clearly detected on all datasets, as previously discussed. More importantly, the high variety on the generated set, observed by the high Intra FID, is witnessed since very early iterations when using $\alpha_{sigm}$, $\alpha_{soft}$, and $\alpha_{tanh}$. The observed mitigation of mode collapse is carried out throughout the whole training.

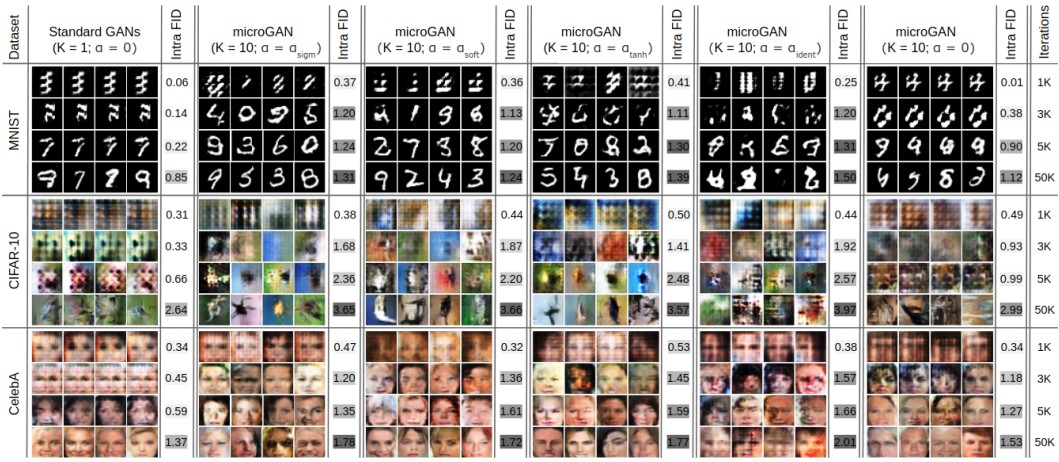

Figure 6: Generated samples from 1K, 2K, 5K and 50K iteration with the respective Intra FID.

When using standard GANs, we notice severe mode collapse, especially early on training. When using 10 discriminators and $\alpha$ set to 0, we notice a slight variation in the generated set, yet, this is only detected after a decent number of iterations, when each $D$ has seen enough samples to guide its judgment to a specific data mode due to the usage of different microbatch for each $D$, delaying sample variety substantially. Thus, using positive $\alpha$ values is shown to be a necessary measure to stimulate variety since the beginning and until the end of training.

## 5 METHOD COMPARISON

We proceeded to compare different settings of microGAN to other existing methods on 3 different datasets: CIFAR-10, STL-10 (Coates et al. (2011)), and ImageNet (Deng et al. (2009)). We downsampled the images of the last two datasets down to 32x32 pixels. We used Inception Score (Salimans et al. (2016)) or IS (higher is better) as the first quantitative metric. Even though IS has been shown to be less correlated with human judgment than FID, most previous works only report results on this metric, making it a useful measure for model comparisons. Out of fairness to the single discriminator methods that we compare our method against, we used only 2 discriminators in our experiments. The architectures and training settings used for all the experiments can be found in the Appendix.

The comparison results are shown in Table 2. We point special attention to the underlined method representing standard GANs, since it was the only method executed with our own implementation and identical training settings as microGAN. Thus, this represents the only method directly comparable to ours. We notice a fair improvement of IS on all the tested datasets, observing an increase up to around 15% for CIFAR-10, 7% for STL-10, and 5% for ImageNet. This indicates the success of our approach on improving the standard GANs framework on multiple datasets with different sizes and challenges.

On CIFAR-10, microGAN achieves competitive results, significantly outperforming GMAN with 5 discriminators while using a similar architecture. We argue that the use of more powerful architectures in the higher ranked methods plays a big role in their end score, especially for DCGAN. Nonetheless, we acknowledge that using different objectives for each $D$ (as proposed in D2GAN) seems to be beneficial in a multi-discriminator setting, representing a good path to follow in the future. Moreover, we observe that using extra autoencoders (DFM) or classifiers (MGAN) in the framework can help to achieve a better performance in the end. However, we note that MGAN makes use of a 10 generator framework, on top of an extra classifier, to achieve the presented results. Furthermore, the generated samples presented in their paper (Hoang et al. (2017)) indicate signs of partial mode collapse, which is not reflected in its high IS.

Table 2: Inception scores. For a fair comparison, only unsupervised methods are compared.

|  | CIFAR-10 | STL-10 | IMAGENET |
|---|---|---|---|
| REAL DATA | 11.24 | 26.08 | 25.78 |
| WGAN (ARJOVSKY ET AL. (2017)) | 3.82 | - | - |
| MIX+WGAN (ARORA ET AL. (2017)) | 4.04 | - | - |
| ALI (DUMOULIN ET AL. (2016)) | 5.34 | - | - |
| BEGAN (BERTHELOT ET AL. (2017)) | 5.62 | - | - |
| MAGAN (WANG ET AL. (2017)) | 5.67 | - | - |
| GMAN (K = 2) (DURUGKAR ET AL. (2016)) | 5.87 | - | - |
| GANs* (GOODFELLOW ET AL. (2014)) | 5.92 | 6.78 | 7.04 |
| GMAN (K = 5) (DURUGKAR ET AL. (2016)) | 6.00 | - | - |
| DCGAN (RADFORD ET AL. (2015)) | 6.40 | 7.54 | 7.89 |
| IMPROVED-GAN (SALIMANS ET AL. (2016)) | 6.86 | - | - |
| D2GAN (NGUYEN ET AL. (2017)) | 7.15 | 7.98 | 8.25 |
| DFM (WARDE-FARLEY AND BENGIO (2016)) | 7.72 | 8.51 | 9.18 |
| MGAN (HOANG ET AL. (2017)) | 8.33 | 9.22 | 9.32 |
| MICROGAN ($K = 2; \alpha = \alpha_{sigm}$) | 6.77 | 7.23 | 7.32 |
| MICROGAN ($K = 2; \alpha = \alpha_{soft}$) | 6.66 | 7.19 | 7.40 |
| MICROGAN ($K = 2; \alpha = \alpha_{tanh}$) | 6.61 | 7.07 | 7.40 |

We further compared our best FID with a subset of the reported methods in Lucic et al. (2017), namely GANs, both with the original and modified objective, LSGAN, and DRAGAN on CIFAR-10 (Table 3). These methods were chosen since they represent interesting variants of standard GANs. We extended each method to an ensemble of discriminators, for a fair comparison to our multiple discriminator approach. We used the same architecture of the last experiment for all methods. We observe that all variants of microGAN outperform the rest of the compared methods under controlled and equal experiments.

Table 3: Minimum FID comparison.

|  | CIFAR-10 |
|---|---|
| GANs (GOODFELLOW ET AL. (2014)) | 70.73 |
| MOD-GANs (GOODFELLOW ET AL. (2014)) | 79.58 |
| LSGAN (MAO ET AL. (2017)) | 83.66 |
| DRAGAN (KODALI ET AL. (2017)) | 80.57 |
| GANs ($K = 2$) | 74.07 |
| MOD-GANs ($K = 2$) | 71.96 |
| LSGAN ($K = 2$) | 73.33 |
| DRAGAN ($K = 2$) | 75.83 |
| MICROGAN ($K = 2; \alpha = \alpha_{sigm}$) | 66.93 |
| MICROGAN ($K = 2; \alpha = \alpha_{soft}$) | 65.54 |
| MICROGAN ($K = 2; \alpha = \alpha_{tanh}$) | 65.84 |

A subset of the generated samples produced by the different variations of microGAN reported in Table 2 are shown in Figure 7, where we observe high variety and realism across all generated sets. Extended results are provided in the Appendix

# 6 CONCLUSIONS

In this work, we present a novel framework, named microGAN, where each $D$ performs microbatch discrimination, differentiating between samples within and outside its fake microbatch. This behavior is enforced by the diversity parameter $\alpha$, that is indirectly self-learned by $G$. In the first iterations, $G$ increases $\alpha$ to lower its loss, expanding its output. Then, as $\alpha$ gradually saturates and each $D$ learns how to better distinguish between real and fake samples, $G$ is forced to fool each $D$ by promoting realism in its output, while keeping the diversity in the generated set. We show evidence that our solution produces realistic and diverse samples on multiple datasets of different sizes and nature, ultimately mitigating mode collapse.

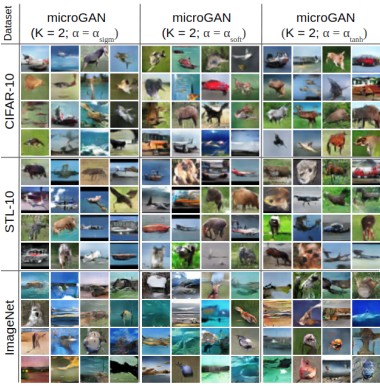

Figure 7: CIFAR-10, STL-10, and ImageNet results.

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

## A  ALGORITHM

The training procedure of microGAN is presented in Algorithm 1.

---

**Algorithm 1** microGAN.

---

**Input:** $K$ number of discriminators, $\alpha$ diversity parameter, $B$ minibatch size
**Initialize:** $m \leftarrow \frac{B}{K}$
**for** number of training iterations **do**
  • Sample minibatch $z_i$, $i = 1 \ldots B$, $z_i \sim p_g(z)$
  • Sample minibatch $x_i$, $i = 1 \ldots B$, $x_i \sim p_r(x)$
  **for** $k = 1$ to $k = K$ **do**
    • Sample microbatch $z_{k_j}$, $j = 1 \ldots m$, $z_{k_j} = z_{(k-1) \times m + 1 : k \times m}$
    • Sample microbatch $x_{k_j}$, $j = 1 \ldots m$, $x_{k_j} = x_{(k-1) \times m + 1 : k \times m}$
    • Sample microbatch $z'_{k_j}$, $j = 1 \ldots m$, $z'_{k_j} \subset z_i \setminus \{z_{k_j}\}$
    • Update $D_k$ by ascending its stochastic gradient:

$$\nabla_{\theta_{D_k}} \frac{1}{m} \sum_{j=1}^{m} [\log D_k(x_{k_j}) + \log(1 - D_k(G(z_{k_j}))) + \alpha \times \log D_k(G(z'_{k_j}))]$$

  **end for**
  • Update $G$ by descending its stochastic gradient:

$$\nabla_{\theta_G} \sum_{k=1}^{K} \Big[ \frac{1}{m} \sum_{j=1}^{m} [\log(1 - D_k(G(z_{k_j}))) + \alpha \times \log D_k(G(z'_{k_j}))] \Big]$$

**end for**

---

## B  THEORETICAL DISCUSSION

To better understand how our approach differs from the original GANs in promoting variety in the generated set, we used a simplified version of the minimax game where we freeze each $D_k$ and train $G$ until convergence. In the most extreme case, we say that we have mode collapse when:

$$\text{For all } z' \sim p_g(z), G(z') = x \tag{4}$$

**Theorem 1.** *In original GANs, mode collapse fully minimizes G's loss when we train G exhaustively without updating D.*

*Proof.* The optimal $x^*$ is the one that maximizes $D$'s output, where:

$$x^* = \underset{x}{\operatorname{argmax}} D(x)$$

Thus, assuming $G$ would eventually learn how to produce $x^*$, mode collapse on $x^*$ would fully minimize its loss, making $x^*$ independent of $z$. □

**Theorem 2.** *In microGAN, assuming $\alpha > 0$, $x \sim p_g$ must be dependent of $z$ for G to fully minimize its loss, eliminating mode collapse when we train G exhaustively without updating any $D_k$.*

*Proof.* The value function between $G$ and each $D_k$ can be expressed as follows:

$$V(D_k, G) = \mathbb{E}_{x \sim p_r}[\log D_k(x)] + \mathbb{E}_{x' \sim p_g}[\log(1 - D_k(x'))] + \alpha \times \mathbb{E}_{x'' \sim p_g}[\log D_k(x'')] \tag{5}$$

To fully minimize its loss in relation to $D_k$, $G$ must find:

$$x' = \operatorname*{argmax}_{x} D_k(x) \text{ and } x'' = \operatorname*{argmin}_{x} D_k(x)$$

which implies

$$D_k(x') \neq D_k(x'') \implies x' \neq x''$$

Thus, generating different outputs for different $z$ is a requirement to fully minimize $G$'s loss regarding each $D_k$. Since we sum all $V(D_k, G)$ to calculate $G$'s final loss, this also applies to overall adversarial set, concluding the proof.

□

## C    TRAINING SETTINGS

The architectural and training settings used in Sections 3, 4, and 5 are presented in Tables 4, 5, and 6, respectively. For the FID comparison on CIFAR-10 and CelebA in Section 5, we used the same architectures as Table 6 but with a batch size of 64 on both datasets, and ran for 78K iterations on CIFAR-10 and 125K iterations on CelebA.

## D    SIGMOID INITIAL VALUE

In Figure 8, we show and discuss the effects of using different $\beta_{sigm}$ on $\alpha_{sigm}$ on the toy dataset, giving more insights regarding the choice of $\beta_{sigm} = -1.8$ mentioned in Section 3.

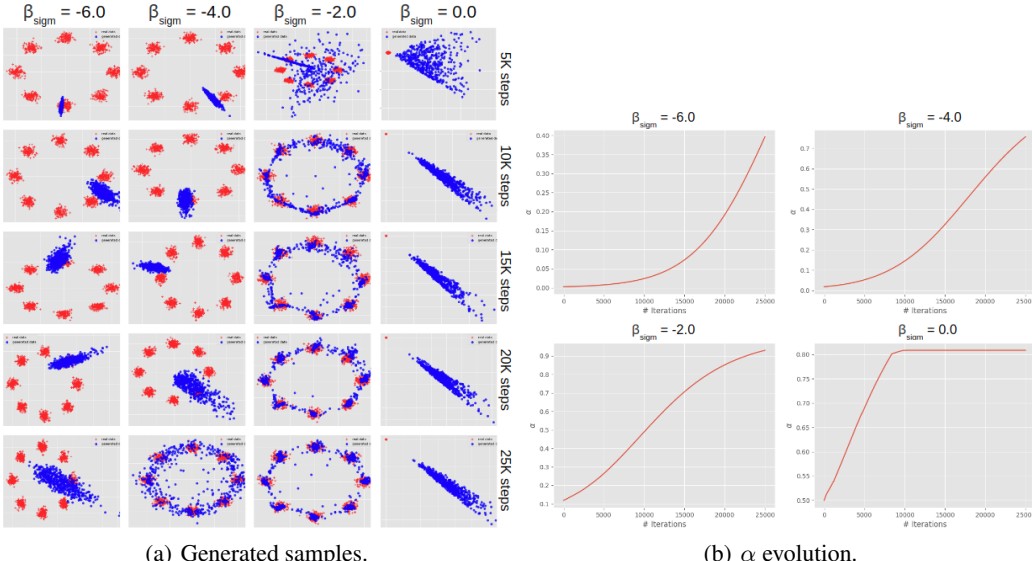

(a) Generated samples.        (b) $\alpha$ evolution.

Figure 8: Analysis of self-learning $\alpha_{sigm}$ with different initial values of $\beta$. The generated samples in (a) show that using lower $\beta_{sigm}$ values lead the model to mode collapse, since only low $\alpha$ values are used throughout the whole training. On the other hand, using higher values, *e.g.*, $\beta_{sigm} = 0.0$, leads to a steeper increase of $\alpha$ values, inducing the model to only generate varied, but not realistic, samples. We empirically found that using $-2.0 \leq \beta_{sigm} \leq -1.8$ led to diverse plus realistic looking samples from early iterations due to the mild, yet meaningful, increase of $\alpha$ throughout training. The evolution of $\alpha$'s values are presented in (b).

# E TOY DATASET COMPARISONS

Figure 9 shows how different methods compare using the above mentioned toy dataset. We compared microGAN's results (K = 8, $\alpha_{sigm}$) to the standard GAN (Goodfellow et al. (2014)), UnrolledGAN (Metz et al. (2016)), D2GAN (Nguyen et al. (2017)), and MGAN (Hoang et al. (2017)). We observe bigger sample diversity with our method, while still approximating the real data distribution.

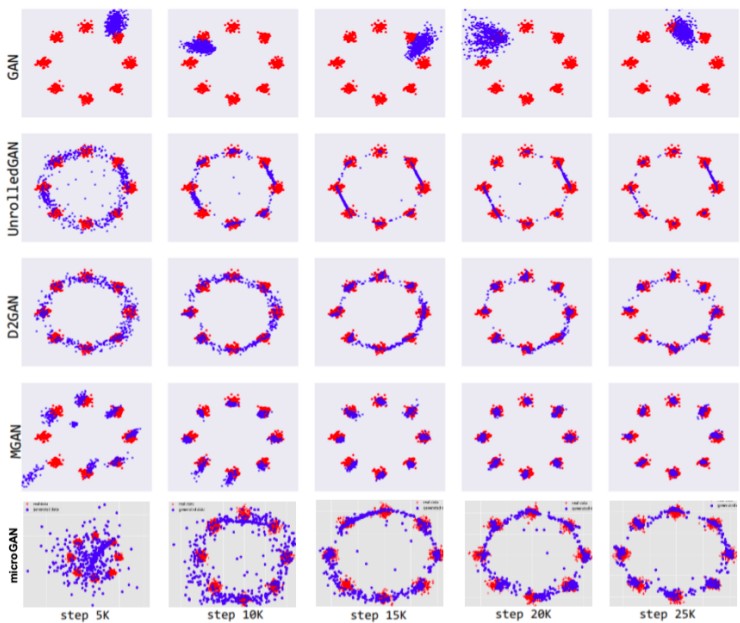

Figure 9: Method comparisons on the toy dataset.

# F EXTENDED RESULTS

Additional results for CIFAR-10, STL-10, and ImageNet are presented bellow.

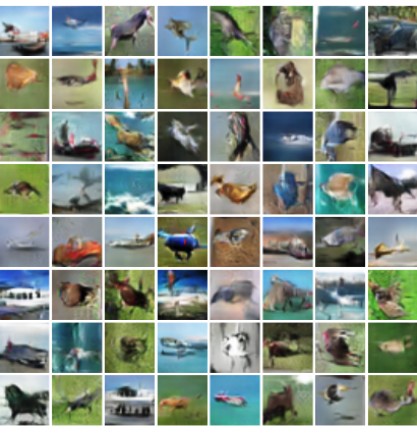

Figure 10: CIFAR-10 extended results using K = 2 and $\alpha_{sigm}$.

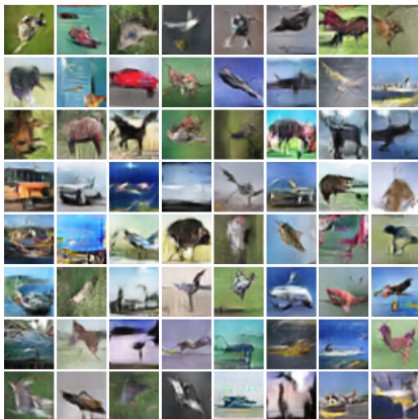

Figure 11: CIFAR-10 extended results using K = 2 and $\alpha_{soft}$.

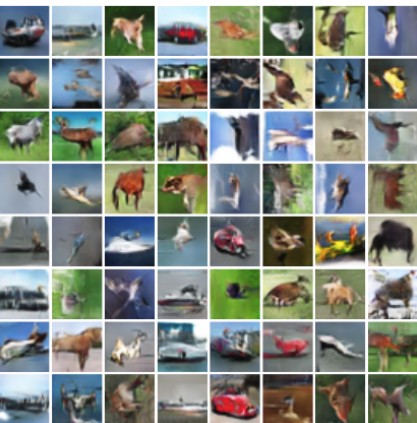

Figure 12: CIFAR-10 extended results using K = 2 and $\alpha_{tanh}$.

Table 4: Training settings for the toy dataset.

|  | FEATURE MAPS | NONLINEARITY |
|---|---|---|
| $G(z) : z \sim Normal(0, I)$ | 256 |  |
| FULLY CONNECTED | 128 | RELU |
| FULLY CONNECTED | 128 | RELU |
| FULLY CONNECTED | 2 | LINEAR |
| $D(x)$ | 2 |  |
| FULLY CONNECTED | 128 | RELU |
| FULLY CONNECTED | 1 | SOFTPLUS |
| NUMBER OF DISCRIMINATORS | 8 |  |
| $\alpha$ (STATIC) | $\{0, 0.1, 0.2, 0.3, 0.4, 0.5, 0.6, 0.7, 0.8, 0.9, 1.0\}$ |  |
| $\alpha$ (SELF-LEARNED) | $\{\alpha_{sigm}, \alpha_{soft}, \alpha_{tanh}, \alpha_{ident}\}$ |  |
| BATCH SIZE | 512 |  |
| ITERATIONS | 25K |  |
| OPTIMIZER | ADAM ($lr = 0.0002, \beta_1 = 0.5$) |  |

Table 5: Training settings for MNIST, CIFAR-10, and CelebA.

| | KERNEL | STRIDES | FEATURE MAPS | BATCH NORM. | NONLINEARITY |
|---|---|---|---|---|---|
| $G(z) : z \sim Uniform[-1, 1]$ | - | - | 100 | - | - |
| TRANSPOSED CONVOLUTION | $3 \times 3$ | $4 \times 4$ | 128 | YES | RELU |
| TRANSPOSED CONVOLUTION | $5 \times 5$ | $2 \times 2$ | 64 | YES | RELU |
| TRANSPOSED CONVOLUTION | $5 \times 5$ | $2 \times 2$ | 32 | YES | RELU |
| TRANSPOSED CONVOLUTION | $5 \times 5$ | $2 \times 2$ | 1/3 | NO | TANH |
| $D(x)$ | - | - | $32 \times 32 \times 1/3$ | - | - |
| CONVOLUTION | $3 \times 3$ | $2 \times 2$ | 32 | YES | LEAKY RELU (0.2) |
| CONVOLUTION | $3 \times 3$ | $2 \times 2$ | 64 | YES | LEAKY RELU (0.2) |
| CONVOLUTION | $3 \times 3$ | $2 \times 2$ | 128 | YES | LEAKY RELU (0.2) |
| FULLY CONNECTED | - | - | 1 | NO | SIGMOID |
| NUMBER OF DISCRIMINATORS | $\{1, 2, 5, 10\}$ | | | | |
| $\alpha$ (STATIC) | $\{0\}$ | | | | |
| $\alpha$ (SELF-LEARNED) | $\{\alpha_{sigm}, \alpha_{soft}, \alpha_{tanh}, \alpha_{ident}\}$ | | | | |
| BATCH SIZE | 100 | | | | |
| ITERATIONS | 50K | | | | |
| OPTIMIZER | ADAM ($lr = 0.0002, \beta_1 = 0.5$) | | | | |

Table 6: Training settings for CIFAR-10, STL-10, and ImageNet.

| | KERNEL | STRIDES | FEATURE MAPS | BATCH NORM. | NONLINEARITY |
|---|---|---|---|---|---|
| $G(z) : z \sim Uniform[-1, 1]$ | - | - | 100 | - | - |
| TRANSPOSED CONVOLUTION | $3 \times 3$ | $4 \times 4$ | 256 | YES | RELU |
| TRANSPOSED CONVOLUTION | $5 \times 5$ | $2 \times 2$ | 128 | YES | RELU |
| TRANSPOSED CONVOLUTION | $5 \times 5$ | $2 \times 2$ | 64 | YES | RELU |
| TRANSPOSED CONVOLUTION | $5 \times 5$ | $2 \times 2$ | 1/3 | NO | TANH |
| $D(x)$ | - | - | $32 \times 32 \times 1$ | - | - |
| CONVOLUTION | $3 \times 3$ | $2 \times 2$ | 64 | YES | LEAKY RELU (0.2) |
| CONVOLUTION | $3 \times 3$ | $2 \times 2$ | 128 | YES | LEAKY RELU (0.2) |
| CONVOLUTION | $3 \times 3$ | $2 \times 2$ | 256 | YES | LEAKY RELU (0.2) |
| FULLY CONNECTED | - | - | 1 | NO | SIGMOID |
| NUMBER OF DISCRIMINATORS | $\{2\}$ | | | | |
| $\alpha$ (SELF-LEARNED) | $\{\alpha_{sigm}, \alpha_{soft}, \alpha_{tanh}\}$ | | | | |
| BATCH SIZE | 100 | | | | |
| ITERATIONS | 200K, 400K, 1M | | | | |
| OPTIMIZER | ADAM ($lr = 0.0002, \beta_1 = 0.5$) | | | | |

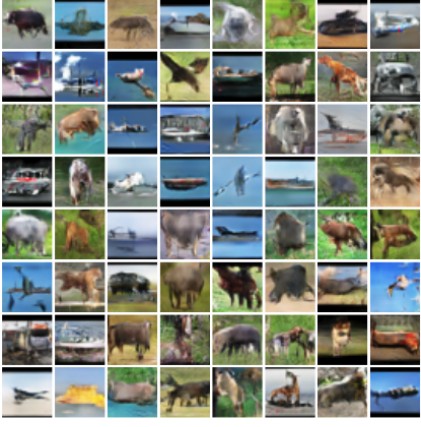

Figure 13: STL-10 extended results using K = 2 and $\alpha_{sigm}$.

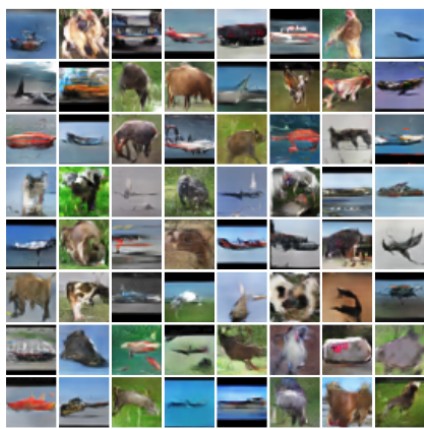

Figure 14: STL-10 extended results using K = 2 and $\alpha_{soft}$.

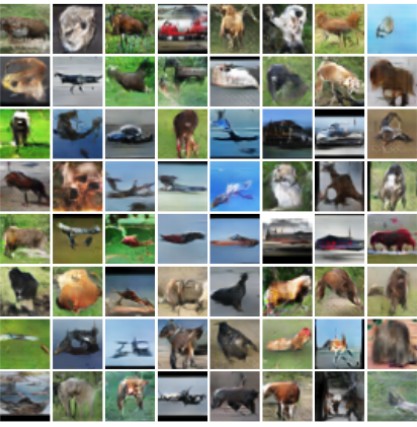

Figure 15: STL-10 extended results using K = 2 and $\alpha_{tanh}$.

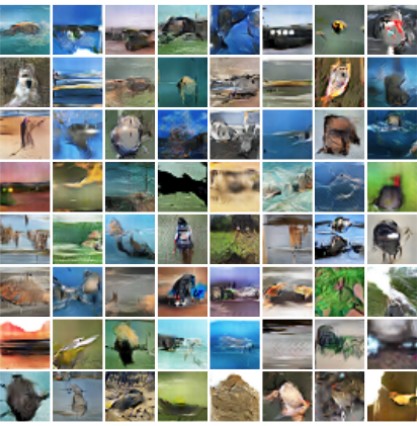

Figure 16: ImageNet extended results using K = 2 and $\alpha_{sigm}$.

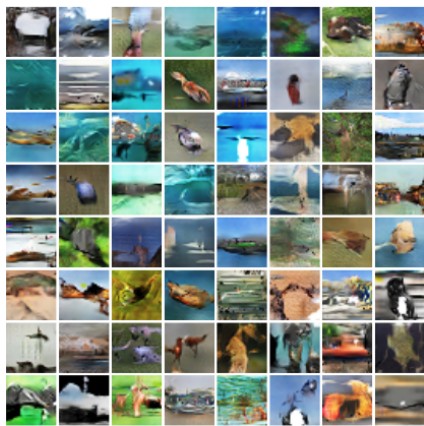

Figure 17: ImageNet extended results using K = 2 and $\alpha_{sigm}$.

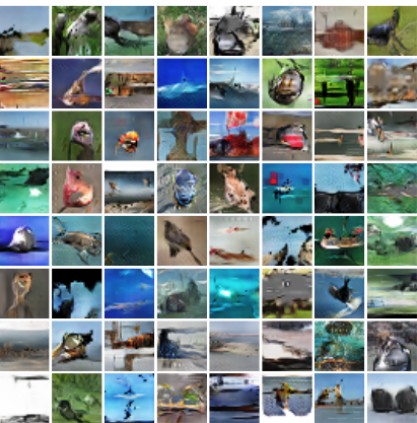

Figure 18: ImageNet extended results using K = 2 and $\alpha_{sigm}$.

