# OpenReview forum: "microGAN: Promoting Variety through Microbatch Discrimination"
_ICLR.cc/2019/Conference_

### Official Review · AnonReviewer3 · 2018-10-29
**Interesting approach; theory and experiments not entirely convincing**

**Rating:** 6
**Confidence:** 3

**Review:**

The submission proposes to increase the variety of generated samples from GANs by a) using an ensemble of discriminators, and b) tasking them with distinguishing not only fake from real samples, but also their fake samples from the fake samples given to the respective other discriminators. The cost function of the minimax GAN optimization problem is changed accordingly. Experimental results suggest that this approach leads to improved results, both visually and w.r.t. FID metric.

Improving on the well known problem of mode collapse in GAN training scenarios is without a doubt an important endeavor. Various methods have been proposed, as curtly summarized in Section 1.1 of the submission.

With respect to the proposed method, I am not completely clear on how it can increase sample variety, if it does. In understand the arguments brought forward in Appendix B, but:
Consider a minimax game involving one generator G and one discriminator D, where each batch of generated samples from G(Z), is split up into two parts, A and B, via selection without replacement using uniform sampling. Z is a matrix of noise inputs, where each column corresponds to one item of the batch. D is now tasked to differentiate whether a sample came from A or B. It seems intuitive to say that in this case, D can neither win, nor provide any useful signal to G, since the sets A and B were split randomly, and there is no influence on G during training. The variety of samples in A as well as in B will be identical to the variety in the set (A and B).

Yet this random microbatch splitting is what seems to be happening here, if I understood Section 3 correctly; just with an ensemble of discriminators, and not just with one.
While it is thus not completely clear to me *why* the proposed additional term seems to bring increased variety, experiments strongly suggest that it does.
As described in Section 3.1, choice of the weighting parameter alpha seems crucial, and additionally alpha needs to depend on the iteration index. Different schedules are demonstrated, but optimality of either is not guaranteed. This makes the actual influence of the additional loss term even harder to judge and evaluate.

Section 4.1 seems to confirm the increase in variety via the self-defined "Intra FID" measure. I would have liked to see this measure evaluated on conventionally trained GANs as a baseline, as well on the methods compared to in Section 5.
In Table 1, both min and mean FID are given over 50k iterations. Instead of reporting the minimum, it might be fairer to compare FIDs after a fixed number of iterations (i.e. 50k in this case).

The method comparison in Section 5 is generally appreciated, but I think some of its flaws are:
- Datasets, including ImageNet, are all downsampled to 32x32 pixels. We have seen generators in recent work that produce interesting high-resolution output in even megapixel size; the tiny size seems like a pessimization of overall approaches.
- The proposed method is compared to other methods using only 2 discriminators, although Section 4.3 suggest a larger number is better.
- MicroGAN does not compare favorably to many of the compared to methods in Table 2. This may not necessarily by a flaw of the MicroGAN contributions, but is rather a problem of an apples-to-oranges comparison, as the authors readily admit ("the use of more powerful architectures [...] plays a big role"). I question the value of such a comparison, if not only the method differ, but also implementation details such as network architectures.

Overall the submission is quite interesting, but not without the above-mentioned flaws.

---

### Official Review · AnonReviewer1 · 2018-11-05
**An apparent flaw in the motivation of the proposed approach undermines confidence**

**Rating:** 3
**Confidence:** 3

**Review:**

The paper proposes a multi-discriminator based extension to GAN training. Specifically, it proposes to split a minibatch of samples into further smaller minibatches (microbatches) and train different discriminators on each. The authors state that "since each D is trained with different fake and real samples, we encourage them to focus on different data properties". This seems incorrect. Random samples drawn from a distribution do not change various statistics of the distribution in expectation (such as means). It only introduces differences due to noise in the sampling process and this noise is not correlated across training iterations or consistent within a microbatch. Without a meaningful/consistent change in the distributions between microbatches there should be no different data properties for the various discriminators to focus in on. As a consequence, a discriminator which evaluates samples independently should not be able to perform this task. Using batchnorm in the discriminator introduces some batch level interactions via the mean and variance statistics but this appears to be serendipity that the authors do not highlight explicitly. As a sanity check - given a fixed generator - if you continue to train the discriminators on randomly drawn samples from this generator distribution does the microbatch discrimination objective continue to make progress and converge to a minimum? What happens if you remove batchnorm so that the samples are processed independently? Is there an additional detail to your paper/method that I missed or misunderstood that addresses the issue raised here? Can you better articulate your intuition on how randomly assigning data to different microbatches results in different data properties? Does the discriminator make use of batch level statistics in some more advanced way beyond just batchnorm such as the minibatch features in Improved GAN (Salimans 2016)?

A baseline of always having alpha set to zero in order to tease out the potential improvements of the proposed approach from the potential benefits of having multiple discriminators would increase confidence in the approach.

Pros:
+ The proposed IntraFID is interesting but is missing two baselines (IntraFID for two batches of real data and IntraFID for two batches of a baseline GAN without the proposed technique) which would help calibrate and contextualize the newly introduced metric.

Cons:
- The paper seems to have a flaw which calls into question whether it is well motivated (see main text).
- The paper does not have any direct/controlled comparisons with other methods that utilize multiple discriminators or batch based discrimination.
- The paper mis-states the Inception Score of Improved GAN. The best result from the Improved GAN paper achieves 6.86 in an unlabeled setting (see -L+HA in the ablation study in Table 3) but is listed as 4.36.
- The paper misses relevant literature - CatGAN (Springenberg 2015) which trains a discriminator to minimize entropy over a categorical distribution assigned to the generator's samples while the generator is trained to maximize entropy of discriminator in this space.  It also misses PACGan (Lin 2017) which also augments a discriminator to look at multiple samples to improve diversity.

---

### Official Review · AnonReviewer2 · 2018-11-05
**Need more work**

**Rating:** 3
**Confidence:** 3

**Review:**

The authors propose a method to improve sample diversity of GANs. They introduce multiple discriminators, each aims to not only compare real and fake examples but also compare different "micro-batch" of examples. The diversity is controlled by a hyperparameter \alpha, and some strategies to set \alpha are studied empirically.

I don't fully understand how the proposed objective can promote diversity. Under what situation will the discriminator fail to discriminate different fake samples?

Many questions are unanswered in the paper.
1. What is the equilibrium of the proposed objective? Does the proposed objective has an equilibrium?
2. How to choose the number of discriminators?
3. How does the proposed approach compare to other papers trying to prevent mode collapse?

The experiment Table 2 and 3 doesn't look convincing to me. The inception score and FID of the proposed approach clearly lack behind state-of-the-art approaches. I don't see any evidence showing that FID of microGAN can be better than, for example, WGAN.

---

### Meta-Review · Area_Chair1 · 2018-12-15
**Needs more thorough justification and evaluation**

**Confidence:** 4
**Recommendation:** Reject

**Metareview:**

The paper proposes an approach to remedying mode collapse problem in GANs. This approach relies on using multiple discriminators and assigning a different portion of each minibatch to each discriminator.

+ preventing mode collapse in GAN training is an important problem

- the exact motivation for the proposed techniques is not fully fleshed out

- the evaluation and baselines used are lacking